# Moderation of Structural DNA Properties by Coupled Dinucleotide Contents in Eukaryotes

**DOI:** 10.3390/genes14030755

**Published:** 2023-03-20

**Authors:** Aaron Sievers, Liane Sauer, Marc Bisch, Jan Sprengel, Michael Hausmann, Georg Hildenbrand

**Affiliations:** 1Kirchhoff Institute for Physics, Heidelberg University, INF 227, 69117 Heidelberg, Germany; aaron.sievers@med.uni-heidelberg.de (A.S.);; 2Institute for Human Genetics, University Hospital Heidelberg, INF 366, 69117 Heidelberg, Germany; 3Faculty of Engeneering, University of Applied Science Aschaffenburg, Würzburger Str. 45, 63743 Aschaffenburg, Germany

**Keywords:** DNA, dinucleotides, *k*-mer, structural DNA properties, 3D conformation, sequence analysis

## Abstract

Dinucleotides are known as determinants for various structural and physiochemical properties of DNA and for binding affinities of proteins to DNA. These properties (e.g., stiffness) and bound proteins (e.g., transcription factors) are known to influence important biological functions, such as transcription regulation and 3D chromatin organization. Accordingly, the question arises of how the considerable variations in dinucleotide contents of eukaryotic chromosomes could still provide consistent DNA properties resulting in similar functions and 3D conformations. In this work, we investigate the hypothesis that coupled dinucleotide contents influence DNA properties in opposite directions to moderate each other’s influences. Analyzing all 2478 chromosomes of 155 eukaryotic species, considering bias from coding sequences and enhancers, we found sets of correlated and anti-correlated dinucleotide contents. Using computational models, we estimated changes of DNA properties resulting from this coupling. We found that especially pure A/T dinucleotides (AA, TT, AT, TA), known to influence histone positioning and AC/GT contents, are relevant moderators and that, e.g., the Roll property, which is known to influence histone affinity of DNA, is preferably moderated. We conclude that dinucleotide contents might indirectly influence transcription and chromatin 3D conformation, via regulation of histone occupancy and/or other mechanisms.

## 1. Introduction

It is a long-known fact that the dinucleotide frequencies in genomic sequences are not random, in the sense that they cannot be reproduced by randomly shuffling nucleotides within a respective sequence [1]. This observation is still true when considering influences of known constraints on longer coding sequences (e.g., amino acid codons). Accordingly, a bias for certain dinucleotide contents exists in sequences without clear annotated functions (e.g., non-coding sequences).

Moreover, it is long known that the transcription of genes is strongly associated with its chromatin state. Specifically, the density of chromatin and therefore histone occupancy have been found to be specific for hetero- and euchromatin [2]. In addition, physical 3D contacts between distant chromatin segments were recently found to be relevant for transcriptional regulation in eukaryotes [3]. The physical bending properties of the chromatin polymer is an important factor that can support or repress the formation of chromatin loops necessary for these contacts, e.g., by shifting the energetic costs to bend the chromatin into a loop configuration. Furthermore, the length and exit angle of linker DNA between histones also seems to play a key role in determining this 3D organization of chromatin [3]. Not bound to histones, the bare linker DNA has a strong negative electric charge. The resulting repulsive forces between DNA segments in combination with the relatively stiff A+T rich sequences, which linker DNA often consist of, lead to very stiff and often intrinsically bent chromatin segments [3]. Taken together, the specific physical properties of linker DNA and influences of histone occupancy on physical properties of chromatin are crucial determinants of 3D chromatin organization and thus transcriptional regulation. Considering the selection for certain dinucleotide contents, it was not a surprise that many different relevant properties (including stiffness) of DNA were found to be connected to dinucleotide frequencies. It was found that dinucleotide context can influence DNA repair [4], DNA bending [5] and hundreds of physical, structural and conformational properties of DNA molecules [6] (DNA properties). Other studies found that these DNA properties were relevant for key functions of DNA, especially the binding of proteins [7], including binding affinities for transcription factors [8,9,10] and histones [11], as well as DNA phase separation [12]. Furthermore, nucleosome occupancy is a determinant of bending properties of chromatin [3,13] and the binding of transcription factors. In addition, transcription factors, such as the transcriptional repressor CTCF, not only regulate the transcription of nearby genes but are also key players in models for chromatin 3D organization (e.g., loop extrusion models [14]). Therefore, besides physical properties of chromatin, the binding of certain proteins, especially transcription factors, are believed to determine functional 3D structures, e.g., so-called topologically associated domains (TADs) [15,16]. Following this deduction, dinucleotide contents seem to influence many factors that determine the 3D conformation of chromatin and transcription regulation. Accordingly, dinucleotide contents might have an indirect influence on these fundamental biological functions. On the other hand, dinucleotide contents between eukaryotic genomes and even between chromosomes within the same genome can differ considerably. If dinucleotides have a considerable influence on structural properties, an equivalent effect on these properties, resulting from different dinucleotide contents, would be expected. Again, these huge differences in DNA properties, e.g., bendability or histone affinity, should result in huge differences in global 3D chromatin conformation. This stays in contrast with the general observation that the principles of 3D organization in eukaryotes are well conserved [17,18], for instance, the presence of TADs (or similar contact domains), chromatin compartments (A/B compartments) and chromosomal territories [19] of comparable size and chromatin density.

Therefore, this observation can only be consistent with a significant influence of structural properties on chromatin 3D organization, if these properties do not change significantly, while the dinucleotide contents determining them can. Appearing as a contradiction, this is actually possible if the influences of some dinucleotides on certain properties are compensated for by the opposite influences of other dinucleotides. Such opposite influences could keep the properties within a range where normal functioning (e.g., formation of organizational 3D structures) is possible and energetically feasible. If, for instance, there was a high content of one dinucleotide that would shift the flexibility of chromatin to a level where the whole chromatin polymer (e.g., chromosome or smaller structure such as TAD) would collapse and therefore could not form a functional 3D conformation, a correlated high (or low) abundance of other dinucleotides could influence the flexibility in the opposite direction to compensate for this potentially hazardous effect. Assuming that the properties needed for normal function of chromatin and the influences of dinucleotides do not change between species, this compensation would require an evolutionary-conserved relation (coupling) of dinucleotide contents between eukaryotic chromosomes, observable as correlated dinucleotide contents.

In this article, we search for these correlations between dinucleotide contents on 2478 chromosomes of 155 eukaryotic species. We analyze the identified correlations to exclude other sources, such as higher abundancies of protein coding sequences (CDS) and genes or regulatory sequences on the respective chromosomes. Finally, we analyze the influence of these correlated dinucleotide contents on physical, structural and conformational DNA properties, using 126 independent predictive in silico models.

## 2. Materials and Methods

### 2.1. k-mer Data, G + C Content and Chromosome Length

Our analysis was performed on *k*-mer data published in [20]. We extended the dataset by applying the same search algorithm and software [20,21] to the genome sequences of 5 additional species to fill some phylogenetic gaps and to therefore reduce associated bias in our data (see Appendix A).

The G + C contents were calculated independently for each chromosome in each genome, by the sum of the contents of respective nucleotides (C and G), based on monomer (k = 1) datasets. The lengths of the respective chromosomes were directly extracted from GenBank files [22] using the Oligo software package [20,21].

### 2.2. Dimer Contents and Normalization

To remove influences of nucleotide contents on the respective dinucleotide frequencies, we normalized the dinucleotide data, using expectation values based on the respective nucleotide contents.

First, we used binomial models to calculate the expectation values E[XY] and variances σ_XY_ for dinucleotide contents for each possible dinucleotide XY (X,Y∈{A,C,G,T}), based on respective nucleotide contents X and Y contents (X,Y∈{A,C,G,T}):(1)$$p_{X}\approx \frac{n_{X}}{L}$$
(2)$$p_{XY}=p_{X}\times p_{Y}$$
(3)$$E\left[XY\right]=Lp_{XY}$$
(4)$$\sigma^{2}=Lp_{XY}\left(1-p_{XY}\right)$$

Here, $n_{X}$ is the count of nucleotides X in the respective sequence (X∈{A,C,G,T}), and L is the length of the sequence (e.g., in nucleotides/bp). We subtracted the expected values from the empirical dinucleotide contents $C_{XY}$ (Equation (5)) to calculate normalized dinucleotide contents ${C^{\prime }}_{XY}$.
(5)$${C^{\prime }}_{XY}=C_{XY}-E\left[XY\right]\pm \sigma$$

We pairwise correlated these normalized dinucleotide contents, interpreting each chromosome as in independent dimension of a vector, using the Pearson correlation coefficient [23], for each possible pair of dinucleotides (128 combinations).

As a reference, we repeated the normalization and correlation with randomly generated dinucleotide datasets. We created these random dinucleotide contents by picking random samples from the binomial distributions described above with p = p_XY_, n = L − 1, independently, 10 times for each chromosome and dinucleotide (over 300,000 independent simulations in total).

To generate an additional reference dataset, we repeated the normalization process, using a second model that directly implements Chargaff’s second rule [24] for dinucleotides, by forcing an equal probability for dinucleotides on different DNA strands. This was implemented by using the average probability of the two for both, the dinucleotide XY in its reverse complement dinucleotide $\hat{XY}$ (X,Y∈{A,C,G,T}).
(6)$$p_{XY}^{C}=p_{\hat{XY}}^{C}=\frac{p_{XY}+p_{\hat{XY}}}{2}$$

Accordingly, for the binomial distribution the samples were taken from, we used p = $p_{XY}^{C}$. Normalization and correlation were repeated a third time using this second model (Chargaff model) (over 30,000 additional independent random samples).

We calculated correlation values r_XY−ZV_ between dinucleotide pairs XY, ZV (X,Y,Z,V∈{A,C,G,T}), using the Pearson correlation coefficient [23]. We derived significance levels $\sigma_{XY-V}$ for the empirical results by dividing the difference between correlation values from empirical data $r_{XY-ZV}^{empirical}$ and the average correlation value from modeled data by the standard deviation from modeled data (for both models independently).
(7)$$\sigma_{XY-ZV}=\frac{r_{XY-ZV}^{empirical}-r_{XY-ZV}^{model}}{\sigma_{XY-ZV}^{model}}$$

### 2.3. Gene Contents and CDS Contents

We extracted the chromosomal contents of genes (including introns) and (protein) coding sequences (CDS) from GenBank files [22], using Oligo [20,21]. To define the gene content, we used the annotations within the GenBank files (genes were explicitly annotated there). We defined CDS as the combination of nucleotides that were part of an mRNA or CDS annotation (default setting of Oligo). The existing CDS annotations within the files alone were not sufficient, since they do not exist in every GenBank file within the dataset. Especially on chromosomes from non-model organisms, CDS annotations are completely absent or incomplete, while mRNA annotations are common and frequent.

For deriving these contents $c_{genes},c_{CDS}$ from nucleotide counts $n_{genes},n_{CDS}$, chromosomal lengths corrected for gaps were used (see Equation (9)).
(8)$$c_{\frac{genes}{CDS}}=\frac{n_{\frac{cds}{genes}}}{L-n_{gaps}}$$

L is the length of the respective chromosome (in nucleotides/bp) and $n_{gaps}$ the count of nucleotides that were part of annotated gaps or annotated as N (unknown nucleotide) in the GenBank files or sequences, respectively. These genes/CDS contents were correlated with normalized dinucleotide contents using the Pearson correlation coefficient. We calculated significance levels for the correlations analogous to Section 2.2 by correlating the gene and CDS contents with dinucleotides sampled from both (random) model distributions as reference and then calculating the differences to the correlation values between genes/CDS and empirical dinucleotide contents (for both models independently).

### 2.4. Enhancer Contents

We downloaded enhancer data from the Enhancer Atlas database [25]. Sufficient data were only available for *Homo sapiens*, *Mus musculus*, *Drosophila melanogaster*, *Gallus gallus*, *Danio rerio*, *C. elegans*, *Rattus norvegicus*, *Saccharomyces cerevisiae* and *Sus scrofa*. Accordingly, our analysis on enhancers was limited to these 9 genomes.

Assuming that the datasets were far from complete and, accordingly, that the relatively high numbers of annotated enhancers in, e.g., *Homo sapiens* and *Mus musculus*, do not reflect actual higher numbers but only higher efforts for searching enhancers in model organisms and species was more relevant for medical research. Therefore, we normalized the enhancer counts per chromosome $c_{Enhancers}$ by the average count of enhancers in the respective genome $\mu_{Enhancers}$, divided by the standard deviation within the respective genome $\sigma_{Enhancers}$:(9)$$c_{Enhancers}^{\prime }=\frac{c_{Enhancers}-\mu_{Enhancers}}{\sigma_{Enhancers}}$$

This method comes with the limitation that we lose any information on higher enhancer counts in genomes, relative to other genomes, that were not an artifact of the higher efforts discussed above. These normalized enhancer counts were correlated with normalized dinucleotide contents using the *Pearson correlation* coefficient (as performed for genes and CDS). We calculated significance levels analogous to Section 2.2 and Section 2.3.

### 2.5. DNA Properties and Models

We downloaded the complete database (126 different dinucleotide models) from the Dinucleotide Properties Genome Browser [6]. These models take the dinucleotide composition of a sequence as input to estimate physical or conformational properties of the associated DNA molecule (DNA properties). Accordingly, we calculated a global/chromosomal mean value for each of the 126 property models based on the empirical dinucleotide data for each chromosome sequence (Equation (10)).
(10)$$V_{E}=\sum_{i,j}M_{ij}c_{ij}$$
where M_ij_ is the property model estimate for the dimer ij, and c_ij_ is the empirical content of ij (i,j ∈ {A,C,G,T}). The contribution of an individual dinucleotide XY (X,Y ∈ {A,C,G,T}) to these values is given by Equation (11).
(11)$$v_{XY}=M_{XY}c_{XY}$$

The contribution of the same amount of random dinucleotides $v_{XY}^{random}$ (based on the binomial dinucleotide models, see Section 2.2 for details) representing the exchange of a dinucleotide by random dinucleotides can be calculated using Equation (12)
(12)$$v_{XY}^{random}=c_{XY}\sum_{ij}M_{ij}p_{ij}^{m}$$

Combining Equations (10)–(12), one can calculate the expected difference $V_{M,AB}^{m}$ of the DNA property if any arbitrary dimer XY would have been removed (−$v_{XY}$) and replaced by random dinucleotides (+$v_{XY}^{random}$) (see Equation (13)).
(13)$$V_{M,XY}^{m}=V_{E}-M_{XY}c_{XY}+c_{XY}\sum_{ij}M_{ij}p_{ij}^{m}$$

For the difference between chromosomal mean values of DNA property, calculated with empirical data only $V_{E}$ (Equation (10)) and with certain dinucleotides exchanged by random dinucleotides XY (X,Y ∈ {A,C,G,T}), $V_{M,XY}^{m}$ (Equation (13)) can then be interpreted as the influence of selection on certain dinucleotide contents on respective DNA properties, e.g., the influence of the overrepresentation of the dinucleotide AT on the roll property of a chromosome.

We calculated this influence $V_{E}-V_{M,XY}^{m}$ for each dinucleotide and property on each chromosome in the dataset. We then calculated mean values over all chromosomes (Appendix A) and divided the results as standard deviation σ to calculate significance levels (Appendix A). A value of >1σ was considered significant.

## 3. Results

### 3.1. Dinucleotide Content Correlations

We normalized the dinucleotide contents and calculated correlation values between every possible pair of dinucleotides, as described in Section 2.1. In general, three classes of results are possible for the pairwise correlation of dinucleotide contents: (1) significant correlation, (2) significant anti-correlation and (3) no significant correlation. We found examples of all three classes (Figure 1, Appendix A).

Applying the Chargaff model (see Section 2.1) as reference, we found significant correlations between the following dinucleotide pairs and their reverse complement pairs: TT-AA, TA-AT, AC-AT, TG-CA, GT-AC, AG/CT-CA/TG, GA/TC-AT, TC-GA, CT-AG, CC/GG-AA/TT, CC/GG-CA/TG, CC/GG-AG/CT, GG-GT, GG-CC, GC-AC/GT, CG-AT, CG-TA, CG-AC/GT, CG-GA/TC and CG-GC. We found a significant anticorrelation between the following dinucleotide pairs and their reverse complement pairs: AT-AA/TT, TA-AA/TT, AC/GT-AA/TT, CACA/TG-AT, CA/TG-TA, AG/CT-AA/TT, AG/CT-AT, AG/CT-TA, AG/CT-AC/GT, GA/TC-AA/TT, GA/TC-TA, GA/TC-CA/TG, GA/TC-AG/CT, CC/GG-AT, CC/GG-TA, CC/GG-AC/GT, CC/GG-GA/TC, GC-AA/TT, GC-AG/CT, GC-CC/GG, CG-CA/TG, CG-AG/CT and CG-CC/GG. We found no significant correlations between the following dinucleotide pairs and their reverse complement pairs: CA/TG-AC/GT, GA/TC-AC/GT, GC-AT, GC-TA, GC-CA/TG, GC-GA/TC and CG-AA/TT (see Appendix A for details). Above, we did not list reverse complement pairs, e.g., we showed AC-AT but not GT-AT, since if there was a significant correlation between a pair of dinucleotides, the reverse complement pairs always showed a similar (significant) correlation (see Appendix A).

A relation similar to the observed similarity between dinucleotide pairs and their reverse complements is not expected for dinucleotides consisting of identical nucleotides in exchanged order (e.g., GC and CG). Such a relation would be observed if the nucleotide content (or G + C content) and not dinucleotide structures are responsible for the observed correlations. As expected, no such relation for dinucleotides with identical nucleotides but reversed order was observed (see Appendix A, Figure 2). For instance, different correlation classes were observed for GC and CG depending on the dinucleotides they were paired with (e.g., GC-CG, GC-CG, GC-AA and CG-AA consistently show no significant correlation (correlation values −0.09 and −0.03, respectively) but GC-AG (correlation value: −0.12) and CG-AG (correlation value: −0.79) belong to different classes). The absence of a relation for dinucleotides consisting of identical nucleotides in exchanged order supports the relevance of analyzing dinucleotide contents and not only nucleotide or G + C contents. The same is true for complementary (but not reversed) sequences, e.g., AC and TG, where no similar relation to reverse complementary dinucleotides was observed (see Appendix A, Figure 2).

### 3.2. Relation to Genes, Coding Sequences (CDS), Enhancers and Chromosome Length

While the results in Section 3.1 confirmed that the observed correlations cannot be a result of nucleotide or G + C contents, it would still be possible that they are a result of more complex but well-known higher-order sequence constraints, such as amino acid codons [26] or sequence patterns of enhancers [27]. Accordingly, we calculated correlation values between normalized dinucleotide contents and chromosomal densities of genes, CDS and enhancers. We also calculated correlation values between normalized dinucleotide contents and chromosomal length, since such a correlation could be an indication for a structural role of dinucleotides. The underlying hypothesis here is that larger chromosomes might form more complex 3D structures. The results are shown in Figure 3 and Table 1 and Appendix A.

Several significant correlations were found. Nearly every dinucleotide correlated with chromosomal length was also anti-correlated with gene and CDS content and vice versa (see Table 1). Accordingly, one can define two major classes of dinucleotides AG, CT, CC, GG, CA, TG, which were correlated with chromosome length and anti-correlated with genes and CDS and GA, TC, AC, GT, GC, CG, which were anti-correlated with chromosome length and correlated with genes and CDS. This observation might be a result of a general correlation between chromosomal length and the amount of non-coding sequences in eukaryotes. In any case, correlations between dinucleotides within these two classes or anti-correlations of dinucleotides from different classes could be the result of the CDS content of respective chromosomes and are therefore arguably a result of a selection on certain amino acid codons (trinucleotides). We observe that dinucleotides and their reverse complement dinucleotides always belong to the same classes described above. Remarkably, dinucleotides not classified (AA, TT, AT, TA) all consist of adenine (A) and thymine (T) nucleotides only. While AA, TT and TA are not correlated or anticorrelated with any tested attribute, AT is correlated with chromosomal length and CDS content of chromosomes (see Table 1). For enhancer counts, the correlations were mostly insignificant with an exception of moderate significance for CA, TG and GC. The reason for the low significance is arguably the relatively small set of analyzed sequences compared to the other correlations (see Section 2.4 for details). Nevertheless, it is remarkable that these dinucleotides were from different classes, while we would have expected a correlation between enhancers and genes.

In Figure 4, the significance levels of correlations between dinucleotide contents is visualized as a heatmap. The dinucleotides were sorted based on the classes identified in Table 1.

If the correlations between dinucleotide pairs observed in Section 3.1 were the indirect result of associations between these dinucleotide contents and sequence constraints of genes or CDS, one would expect correlations between dinucleotide contents within the same class and anti-correlations between dinucleotide contents from different classes. This association is observed for most but not all significantly correlated dinucleotide pairs (Figure 4). Significantly correlated pairs not associated with genes/CDS/length classification are listed in Table 2.

Consistent with Figure 4, many listed dinucleotide pairs include at least one dinucleotide consisting of A and T only. Since no correlations between these dinucleotides and genes, CDS or length were observed, they were not classified and, accordingly, neither of their correlations with other dinucleotides observed in Section 3.1 are associated with their classification. Pairs without such A/T-only dinucleotides listed are GC-CA, GC-TG, GC-GA, GC-TC and CG-GC. These dinucleotide pairs were of particular interest, since their correlations were unexpected considering the classification in Table 1, and therefore, they are the result of a bias with yet unknown origin.

### 3.3. Structural DNA Properties

We calculated the influence of dinucleotide contents on physical and conformational DNA properties for all chromosomes analyzed (see Section 2.5 for details on the calculation). We found that many dinucleotides have significant influences on DNA properties (see Appendix A). While significant, in a statistical sense, most changes were small compared to the original values (see Appendix A). While we cannot exclude that even small changes on physical or structural DNA properties might have considerable effects on chromatin conformation or function, we concluded that larger changes on DNA properties will most likely have larger effects on DNA conformation and function. Therefore, we focused on significant DNA property changes larger than 10% of the original value (see Figure 5, Table 3).

Some DNA properties are listed multiple times in Figure 5 and Table 3. This is possible since we used multiple models (with different IDs) for predicting the same DNA property in some cases. The fact that these properties are listed multiple times can therefore be interpreted as multiple independent models making the same consistent, thus more reliable, prediction of a significant and large change of a DNA property resulting from changes in dinucleotide composition. DNA properties listed multiple times are roll, tilt and slide.

While in most cases, dinucleotides and their reverse complements influence the same DNA properties in the same direction significantly, the direction property is one case where the GG dinucleotide content gives a positive contribution, while CC contributes negatively. Accordingly, in this case, a moderation of the direction property is the direct result of Chargaff’s second law [24].

### 3.4. Influence of Correlated Dinucleotide Pairs on DNA properties

We combined the results from Section 3.1, Section 3.2 and Section 3.3 by searching for pairs of correlated and anticorrelated dinucleotide contents, not associated with genes/CDS classification (see Table 2), that significantly influence DNA properties in the opposite or same direction, respectively (see Table 3).

We found eight significantly correlated dinucleotide pairs, where both correlated dinucleotide contents have a significant influence on the same DNA property in opposite directions, and we found 19 significantly anti-correlated dinucleotide pairs where both correlated dinucleotide contents have a significant influence on the same DNA property in the same direction (see Table 4). In both cases, the coupled (correlated or anticorrelated) dinucleotide contents compensate for each other’s influences on certain DNA properties, therefore moderating these properties and preventing potentially harmful, extreme mean values of these DNA properties over analyzed chromosomes.

Several DNA properties were found to be influenced multiple times by coupled dinucleotide pairs (roll, tilt and twist), indicating that multiple dinucleotide pairs influence the same property significantly. Accordingly, we could not only confirm that dinucleotide pairs moderate DNA properties significantly, but even larger coupled sets (of more than two) of dinucleotides might moderate them.

We observed some interesting patterns in dinucleotide contents influencing certain DNA properties. Twist is only influenced by pairs containing AT and AG/TG and their reverse complements, with negative correlation. Tilt is only influenced by pairs containing TA, AA or TT with AT, AC, AG or their reverse complements. The roll property, which is represented by two independent models, is outstanding for being influenced by pairs where none of the two dinucleotide contents consist of A/T only (see green background in Table 4). This is only observed for roll and direction and is only observed for pairs with GC dinucleotide content. In addition, toll and tilt were both mainly influenced by negative correlated pairs including AA/TT and AC/GT and positive correlations of AC/GT with TA. In both cases, AA/TT and TA influence the property in different directions. The main difference between the patterns observable in Table 4 for roll and tilt is that roll is influenced by pairs including GC, while tilt is influenced by pairs including AT instead. In both cases, the reversed dinucleotide contents (TA and CG) were not relevant for the patterns. Considering the similarities between the patterns of roll and tilt in Table 4, the shared property of AT and GC, that they are identical to their reverse complements, might be of special interest for the moderation of these DNA properties. This may indicate a special role of sequence differences between DNA strands or more explicitly of *Chargaff* rules [24] or their violation. In Table 4, for dinucleotides not consisting of A/T only, AC/GT is by far the most frequent partner influencing dinucleotide properties. This might indicate a special role of AC/GT for the moderation of DNA properties.

## 4. Discussion

We started with the hypothesis of evolutionary coupled dinucleotide contents that moderate DNA properties to support or determine functional chromatin organization. The first prediction derived from this hypothesis is the existence of correlations and/or anticorrelations between dinucleotide contents on eukaryotic chromosomes. We found the expected correlations for a large number of dinucleotide pairs (Figure 1 and Figure 2, Appendix A). Since sequence constraints from known functional elements larger than one nucleotide, not necessarily relevant for physical or structural DNA properties, could also explain the observed correlations, we checked for correlations between dinucleotides and the abundancies of genes, coding sequences (CDS) and enhancers. We found that many of the observed correlations between dinucleotide contents could be the result of associated constraints, while a considerable number of correlated and anticorrelated dinucleotide pairs remained without such explanation (see Figure 3 and Figure 4, Table 1). In general, a correlation with these CDS, genes or enhancers does not exclude a considerable influence of the dinucleotide pairs on DNA properties. Actually, certain DNA properties were found to be predictive for regulatory sequences [10]. Therefore, a correlation between associated dinucleotides and enhancers or genes could still be consistent with our hypothesis on the role of dinucleotide coupling. Since our analysis is unable to distinguish between such correlations as a result of DNA properties and correlations resulting from other sequence constraints (independent of DNA properties) in these regions, we decided to exclude all corresponding dinucleotide pairs from the downstream analysis. This rather conservative filtering prevents false positive results as a consequence of sequence constraints on CDS, genes or enhancers. We also checked correlations with the length of the respective chromosomes. Since larger chromosomes could potentially form larger, higher numbers or more complex functional large-scale 3D chromatin structures (e.g., TADs), a correlation of dinucleotides with chromosomal length could give a first hint on their relevance for 3D chromatin organization. We found that most dinucleotides correlated with chromosomal length were anticorrelated with CDS and genes and vice versa (see Table 1), which allows for binary classification of dinucleotides based on correlations with CDS/genes and anti-correlated with chromosomal length. While a systematic analysis is out of the scope of this article, the observation could indicate that the content of intergenic sequences (formerly sometimes falsely referred to as junk DNA [28]) is correlated with chromosomal length in eukaryotes. Following the argumentation above, this could be a hint that large stretches of intergenic sequences were key players on chromatin 3D organization, e.g., as simple spacer elements or by providing certain chromatin properties (e.g., physical properties or protein binding affinities). This would be an additional hypothesis independent but compatible with the hypothesis on dinucleotide coupling discussed here.

The only dinucleotide content found to be correlated with chromosomal length not anti-correlated with genes and CDS is AT content (see Table 1). Since we corrected the contents considering influences of nucleotide contents, this correlation cannot be explained by the lower G + C content of intergenic sequences. In addition, a correlation between AT and CDS was observed, while CDS was known to be G + C rich compared to intergenic regions. While our method cannot provide clear evidence on the function underlying unexpected correlations, one might consider these observations as a hint toward a potential role of AT-mediated DNA properties potentially influencing 3D chromatin organization. The general absence of pure A/T dinucleotides within the presented classification scheme (AA, TT and TA are not correlated with CDS or chromosome length, and therefore, their presence cannot be explained by associated sequence constraints) could also hint toward a special functional role of those. Since poly-A stretches were known to be determinants of histone positioning [29,30], and histone occupancy is a determinant of chromatin flexibility [3], which is believed to be relevant for chromatin 3D organization, this role could be the maintenance or support of certain 3D configurations by providing beneficial chromatin flexibility, e.g., reducing associated energetic costs for the formation of functional chromatin configurations. We found additional support of this hypothesis by the observation that many dinucleotide correlations including pure A/T dinucleotides were found to balance the roll property (see Table 4) while the roll property was found to be relevant for histone affinity of DNA [11]. Accordingly, a mediation of histone occupancy as a result of correlation of dinucleotide contents would be plausible. On the other hand, AT dinucleotides were known to be relatively flexible (increasing bendability of DNA) [30,31,32], and AT is the only dinucleotide consisting of only A/T, influencing the roll property in the opposite direction of AA [6]. In combination, the hypothetical function of AA/TT and AT could be the encoding of histone affinity and DNA bendability, affecting energetic costs of functional chromatin conformations (e.g., TADs or other loops).

Independent of correlated dinucleotide pairs, we also found significant influence of individual dinucleotide contents on many DNA properties (see Appendix A) besides roll. This observation supports the general importance of dinucleotide contents on DNA properties on chromosomal scales. While direct models for the flexibility of DNA were part of our analysis (see e.g., persistence length (15) in Appendix A), they were not found to be influenced significantly by dinucleotide contents. Therefore, a direct connection between dinucleotide-modulated DNA flexibility and the observed non-random distribution of dinucleotides is not supported by our results. Although, we cannot exclude significant effects below the introduced threshold of relative changes of more than 10% of the original chromosomal average. Accordingly, 21 DNA properties were further analyzed (see Table 3), including four different models for the roll property. In combination with the outstanding role of the roll property [11] for histone affinity and thus chromatin bendability, the consistent prediction of significant effects of dinucleotide contents on four independent models underlines their relevance for chromatin 3D organization. Not only pure A/T dinucleotides (AA, TT, AT, TA) were found to influence the roll property considerably (see Table 3). CG content is the only dinucleotide content not listed to influence the roll property significantly and for more than 10% in at least one model’s predictions. This could indicate that the roll property in general is very sensitive to changes in dinucleotide contents.

The main prediction derived from the hypothesis of coupled dinucleotide contents moderating DNA properties to support or determine functional chromatin organization is the influence of certain DNA properties by pairs (or larger groups) of dinucleotide contents in different directions to “balance” these properties to a somehow “moderate” value. Confirming this prediction, we found 27 pairs of dinucleotide pairs significantly influencing DNA properties in opposite directions (Table 4), therefore balancing these DNA properties in the expected way. We observed that different dinucleotide contents seem to moderate specific DNA properties. The twist property is only moderated by pairs containing AT paired with AG/TG (or their reverse complements) while tilt is only moderated by TA, AA and TT paired with AT, AC, AG (or their reverse complements). Roll is moderated by pairs containing at least one pure A/T dinucleotide (AA, TT, AT, TA) or the GC dinucleotide content. The occurrence of AA, TT and GC is of special interest, since they were known to result in curved but relatively stiff DNA molecules [31,32]. Additionally, the stiffness and curvature of AA and TT were known to reduce histone affinity, which is consistent with the observed moderation of the roll property by these dinucleotide contents, since the roll property was found to predict nucleosome occupancy [11]. This supports the potential relevance of dinucleotide contents for 3D chromatin organization.

As we already mentioned, we used quite conservative filters, excluding dinucleotides correlated with CDS/genes and excluding significant (in a statistical sense) changes less than 10% of the original values. Therefore, there might be an even larger number of unanalyzed dinucleotide pairs moderating DNA properties, relevant for 3D chromatin organization or other functions (e.g., protein or complex binding affinities apart from histones). Additionally, this work focused on DNA property changes on chromosomal scales, while influences of dinucleotide contents or pairs of such could also influence functions on smaller length scales, ranging from the level of individual TADs to local surroundings of individual genes, enhancers or other functional elements. For instance, we found no direct influence of dinucleotide content on the DNA flexibility on a chromosomal scale, while we would still expect such an influence on a local scale (e.g., in linker DNA). The chromosomal scale observations in this work might only be the tip of the iceberg for influences of DNA properties and functions by coupled dinucleotide properties that are yet to be discovered.

In summary, we conclude that the DNA sequence, especially coupled dinucleotide contents, might be a yet overseen possible determinant controlling chromatin organization on the nanoscale. As passive elements incorporated into DNA sequences, dinucleotides may have become relevant players for 3D chromatin folding and spatial organization. Assuming that further studies support our hypothesis, dinucleotide contents might have strong impact on nucleosome positioning and accordingly on inter-nucleosomal potentials [33], thus indirectly influencing 3D chromatin organization, together with epigenetic interactions, to form a powerful control system for genome functioning [34].

## Figures and Tables

**Figure 1 genes-14-00755-f001:**
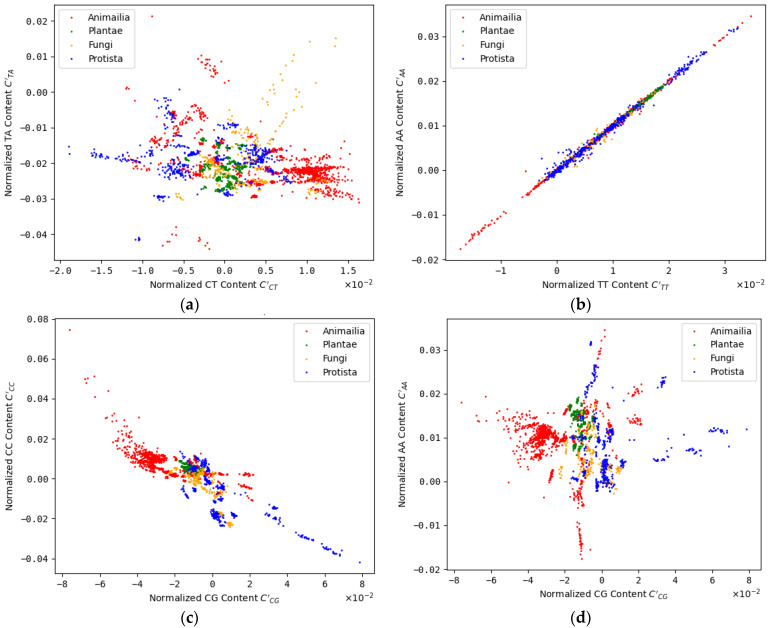
Examples for dinucleotide relations: Representative collection of possible relations between normalized dinucleotide contents (see Section 2.1) on all eukaryotic chromosomes analyzed. (**a**) Moderate anti-correlation between normalized CT and AT contents (correlation value: −0.51), (**b**) high correlation between normalized TT and AA contents (correlation value: 0.997), (**c**) high anti-correlation between normalized CG and CC contents (correlation value: −0.82), (**d**) no significant correlation between normalized CG and AA contents (correlation value: −0.035).

**Figure 2 genes-14-00755-f002:**
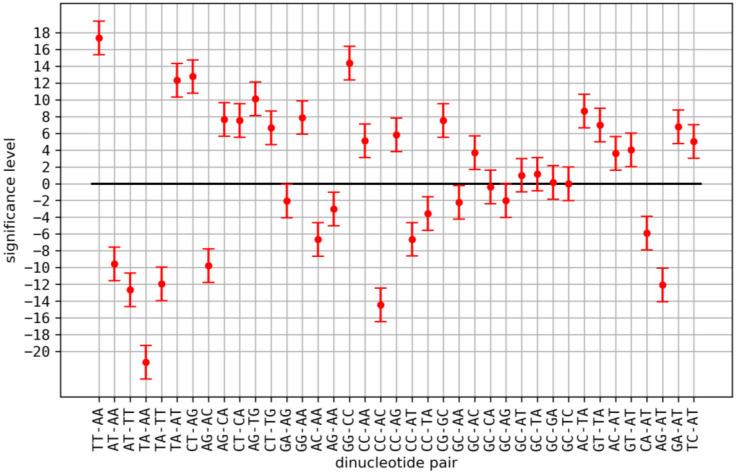
Significance levels of dimer pairs: a representative selection (see Appendix A for all dinucleotides) of significance levels for correlations (see Section 2.1) of different dinucleotide pairs (x-axis) in relation to the *Chargaff* model. Deviations from zero larger than 1.0 were not expected by the model and are thus considered significant.

**Figure 3 genes-14-00755-f003:**
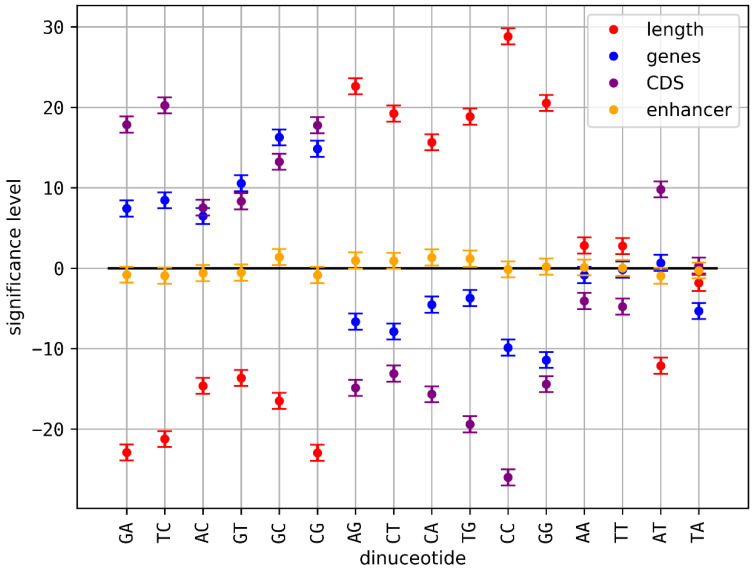
Significance levels of dinucleotide-attribute correlations: normalized dinucleotide contents were correlated with different attributes (gene content, CDS content, normalized enhancer count and chromosomal length (in bp)). Significance levels were calculated in relation to (random) Chargaff models samples (see Section 2.1). Values deviating more than ±1.0 from zero are not expected by the model and are thus considered significant.

**Figure 4 genes-14-00755-f004:**
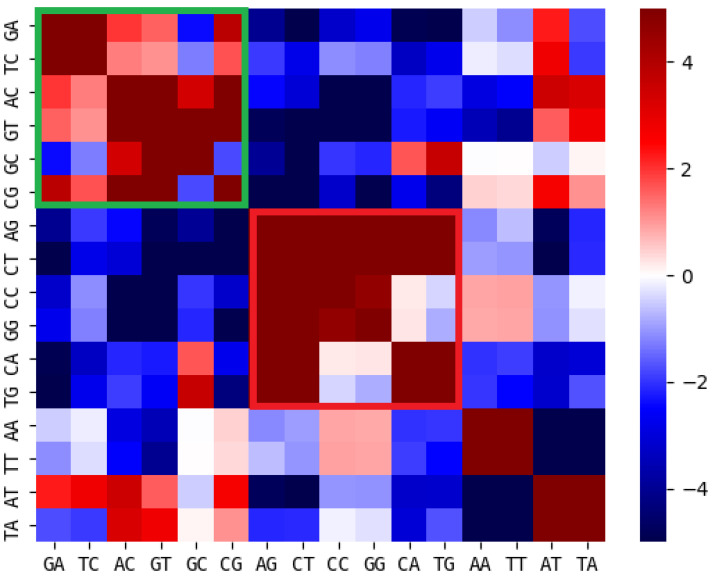
Significance levels of dinucleotide correlations sorted by CDS classes: significance levels of correlations between dinucleotide contents in relation to Chargaff model, sorted by the classes identified using correlations with chromosome length, gene content and CDS content. Correlations within classes are highlighted (green: correlated with genes, CDS anti-correlated with length; red: anti-correlated with genes, CDS correlated with length).

**Figure 5 genes-14-00755-f005:**
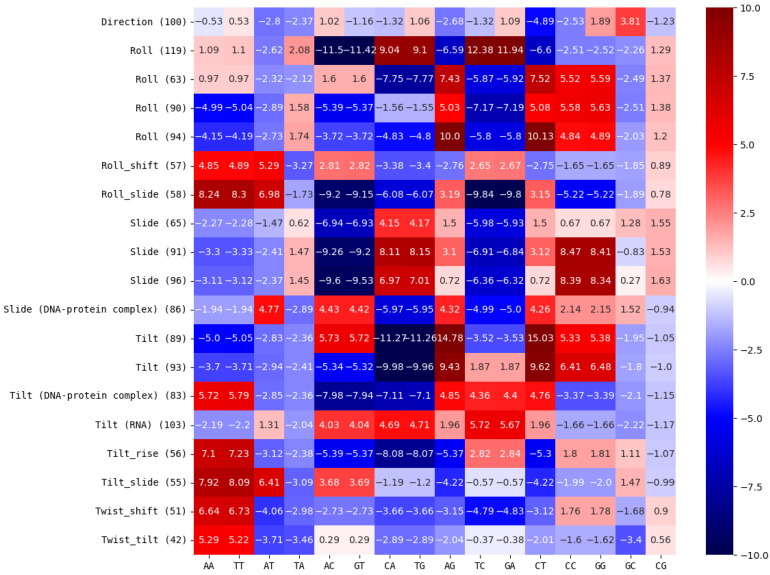
Significance levels of dinucleotide content contributions to DNA properties: heatmap of significance level contributions from dinucleotides to physical/conformational/structural DNA properties (see Section 2.5 for details on calculation). The DNA property model IDs in brackets and DNA property names were taken from the dinucleotide properties database [6]. Different IDs for the same property name represent independent models for the same property. DNA properties without any significantly contributing dinucleotides leading to changes of at least 10% are not listed (see Appendix A). Contributions with significance levels above >1σ were considered significant (see Appendix A).

**Table 1 genes-14-00755-t001:** Dinucleotides correlated or anti-correlated with attributes: lists of dinucleotides significantly correlated or anti-correlated with chromosome length, gene content, CDS content or enhancer count (see Section 2.3, Figure 3, Appendix A). Re-occurring sets of dinucleotides are highlighted with different background colors.

Attribute	Correlated Dinucleotides	Anti-Correlated Dinucleotides
Chromosome length	AG CT CC GG CA TG AT	GA TC AC GT GC CG
Gene content	GA TC AC GT GC CG	AG CT CC GG
CDS content	GA TC AC GT GC CG AT	AG CT CC GG CA TG
Enhancer count	CA TG GC	-

**Table 2 genes-14-00755-t002:** Correlated dinucleotide pairs not explained by genes, CDS or chromosome length correlations. Significantly correlated dinucleotide pairs (see Appendix A). Pairs with correlations expected by a shared significant correlation of both dinucleotides with genes, CDS or chromosome length were filtered.

Dinucleotides		Empirical Correlation Value	Model Correlation Value	Model Significance Level
TT	AA	1.00	0.37 ± 0.10	6.26
AT	AA	−0.56	0.01 ± 0.10	−5.66
AT	TT	−0.56	0.02 ± 0.10	−5.86
TA	AA	−0.63	−0.00 ± 0.11	−5.53
TA	TT	−0.64	0.01 ± 0.10	−6.47
TA	AT	0.69	0.05 ± 0.13	4.98
AC	AA	−0.36	0.04 ± 0.14	−2.89
AC	TT	−0.36	−0.04 ± 0.13	−2.51
AC	AT	0.21	−0.07 ± 0.08	3.49
AC	TA	0.34	−0.02 ± 0.11	3.25
CA	AA	−0.18	0.05 ± 0.11	−2.00
CA	TT	−0.18	0.05 ± 0.12	−1.88
CA	AT	−0.44	−0.05 ± 0.12	−3.24
CA	TA	−0.31	−0.02 ± 0.09	−3.07
TG	AA	−0.18	0.08 ± 0.13	−1.97
TG	TT	−0.18	0.06 ± 0.10	−2.46
TG	AT	−0.44	−0.06 ± 0.12	−3.19
TG	TA	−0.30	−0.06 ± 0.14	−1.69
GT	AA	−0.36	0.04 ± 0.11	−3.49
GT	TT	−0.36	0.05 ± 0.10	−4.03
GT	AT	0.21	−0.01 ± 0.14	1.59
GT	TA	0.34	−0.03 ± 0.13	2.78
AG	AA	−0.15	−0.01 ± 0.12	−1.15
AG	AT	−0.53	0.04 ± 0.12	−4.77
AG	TA	−0.28	0.02 ± 0.14	−2.14
GA	TT	−0.10	−0.02 ± 0.07	−1.10
GA	AT	0.27	0.05 ± 0.10	2.24
GA	TA	−0.19	0.00 ± 0.11	−1.73
TC	AT	0.27	−0.02 ± 0.11	2.78
TC	TA	−0.19	0.05 ± 0.12	−1.94
CT	TT	−0.15	−0.04 ± 0.10	−1.04
CT	AT	−0.53	0.07 ± 0.10	−6.12
CT	TA	−0.28	−0.00 ± 0.13	−2.10
CC	AT	−0.26	−0.09 ± 0.17	−1.03
GG	AT	−0.26	−0.10 ± 0.15	−1.08
GC	CA	−0.02	−0.28 ± 0.16	1.67
GC	TG	−0.01	−0.35 ± 0.09	3.62
GC	GA	0.01	0.30 ± 0.12	−2.41
GC	TC	0.01	0.29 ± 0.21	−1.29
CG	AT	0.49	0.09 ± 0.15	2.66
CG	TA	0.26	0.08 ± 0.16	1.09
CG	GC	0.43	0.54 ± 0.06	−1.78

**Table 3 genes-14-00755-t003:** Dinucleotides with significant contributions to DNA properties. Dinucleotides with contents significantly contributing to physical or conformational DNA properties (see Section 2.5 for details on calculation). The DNA property model IDs and DNA property names were taken from the were taken from the dinucleotide properties database [6]. Different IDs for the same property name represent different models for the same property. Dinucleotide leading to absolute changes of the property < 10% are not listed. DNA properties without any significantly contributing dinucleotides leading to changes larger the 10% are not listed (see Appendix A). Contributions with significance levels above >1σ were considered significant (see Figure 5 or Appendix A).

DNA Property Model ID	DNA Property Name	Dinucleotides with Positive Contribution	Dinucleotides with Negative Contribution
100	Direction	GG GC	CA GT TC CC
103	Tilt (RNA)	GA TC	AA TT
119	Roll	CA TG	AT
42	Twist_tilt	AA TT	AG CT
49	Shift_rise	AA TT CC GG	CA TG
5	Tip	TA	CA TG
51	Twist_shift	AA TT	AT CA TG AG CT
55	Tilt_slide	AA TT AT AC GT	TA AG CT
56	Tilt_rise	AA TT	
57	Roll_shift	AA TT AC GT	TA AG CT
58	Roll_slide	AT	CA TG
63	Roll	AG CT	CA TG GC
65	Slide	CA TG	AC GT
83	Tilt (DNA–protein complex)	AA TT	CA TG
86	Slide (DNA–protein complex)	AT AC GT	CA TG GA TC
89	Tilt	AG CT	CA TG
90	Roll	CC GG	GC
91	Slide	CA TG	
93	Tilt	AG CT CC GG	CA TG
94	Roll	AG CT CC GG	GA TC GC
96	Slide	CA TG	

**Table 4 genes-14-00755-t004:** Correlated dinucleotide pairs significantly influencing DNA properties, not explained by CDS. Dinucleotide pairs with significant correlation (positive sign of significance level) or significant anticorrelation (positive sign of significance level) and significantly changed DNA properties. The significance and sign of the influence of the respective dinucleotides are also listed. Only correlated pairs with opposite influences on DNA properties and anticorrelated pairs with the same influence on DNA properties that change respective properties by at least 10% are included. Rows representing interesting dinucleotide patterns are grouped and marked by different background colors. Green: pairs including a non-pure A/T dinucleotide and GC; Red: pairs including AC/GT with poly-A/T (always negative correlation). Orange: pairs including AG/CT (always negative correlation) with AT. Yellow: pairs including AC/GT (always positive correlation) with TA. Blue: pairs including AA/TT/TA with AT.

Dinucleotide Pair	Significance Level Correlation	DNA Property (ID)	Significance Levels of DNA Property Changes
GA-GC	−2.4	Roll (94)	−5.8 (GA)/−2.0 (GC)
TC-GC	−1.3	Roll (94)	−5.8 (TC)/−2.0 (GC)
AC-AA	−2.9	Roll (57)	+2.8 (AC)/+4.8 (AA)
GT-AA	−3.5	Roll (57)	+2.8 (GT)/+4.8 (AA)
AC-TT	−2.5	Roll (57)	+2.8 (AC)/+4.9 (TT)
GT-TT	−4.0	Roll (57)	+2.8 (GT)/+4.9 (TT)
AC-TA	+3.2	Roll (57)	+2.8 (AC)/−3.3 (TA)
GT-TA	+2.8	Roll (57)	+2.8 (GT)/−3.3 (TA)
CT-TA	−3.1	Roll (57)	−2.8 (CT)/−3.3 (TA)
AA-AT	−5.7	Tilt (55)	+7.9 (AA)/+6.4 (AT)
TT-AT	−5.9	Tilt (55)	+8.1 (TT)/+6.4 (AT)
TA-AT	+5.0	Tilt (55)	+6.4 (AT)/−3.1 (TA)
AC-AA	−2.9	Tilt (55)	+3.7 (AC)/+7.9 (AA)
GT-AA	−3.5	Tilt (55)	+3.7 (GT)/+7.9 (AA)
AC-TT	−2.5	Tilt (55)	+3.7 (AC)/+8.1 (TT)
GT-TT	−4.0	Tilt (55)	+3.7 (GT)/+8.1 (TT)
AC-TA	+3.2	Tilt (55)	+3.7 (AC)/−3.7 (TA)
GT-TA	+2.8	Tilt (55)	+3.7 (GT)/−3.7 (TA)
AG-TA	−2.1	Tilt (55)	−4.2 (AG)/−3.7 (TA)
CT-TA	−3.1	Tilt (55)	−4.2 (CT)/−3.7 (TA)
AG-AT	−4.8	Twist (51)	−3.1 (AG)/−4.1 (AT)
CT-AT	−6.0	Twist (51)	−3.1 (CT)/−4.1 (AT)
CA-AT	−3.2	Twist (51)	−3.7 (CA)/−4.1 (AT)
TG-AT	−3.2	Twist (51)	−3.7 (TG)/−4.1 (AT)
GA-AT	+2.2	Slide (86)	−5.0 (GA)/+4.8 (AT)
TC-AT	+2.8	Slide (86)	−5.0 (TC)/+4.8 (AT)
CA-GC	+1.7	Direction (100)	+3.8 (GC)/−1.3 (CA)

## Data Availability

Our analysis was performed on data we already published in [20], based on data from publicly available databases (GenBank [22]). We extended the dataset with genome sequences of five additional species to fill some phylogenetic gaps (see Appendix A for accession numbers), using GenBank [22]. All codes (python and C), including visualization, used for this article, as well as a manual, are available online at “http://www.kip.uni-heidelberg.de/biophysik/software” or from an associated GitHub repository (https://github.com/Sievers-A/Oligo. Accessed on 7 September 2022). The software was developed under Windows operating systems but can also be used on Unix-based systems (e.g., Linux).

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
