# Peer review of "Moderation of Structural DNA Properties by Coupled Dinucleotide Contents in Eukaryotes"

_genes, 2023, doi:10.3390/genes14030755_

Round 1

Reviewer 1 Report

In the manuscript "Moderation of Structural DNA Properties by Coupled Dinucleotide Contents in Eukaryotes" Aaron et al hypothesized that coupled dinucleotide contents moderate DNA properties to determine functional chromatin organization and they observed considerable correlations between dinucleotide contents on eukaryotic chromosomes. The manuscript may be interesting to researchers in the field of DNA structural biology or computational genomics.

This reviewer found this manuscript premature to be published and efforts are needed to improve it.

Comments:

1. Manuscript is very hard to understand, throughout the manuscript Authors used long sentences which are hard to grasp the idea.

For instance the following sentence (Page 16, Line number 510) "Additionally, the stiffness of AA and TT is known to reduce histone affinity and therefore the chromatin composition considerably which is consistent with the observed moderation of the Roll property by these dinucleotide contents, since the Roll property was found to predict nucleosome occupancy [9] and thus supports the potential relevance of dinucleotide contents for 3D chromatin organization." does not make any sense. AA and TT periodicity play role in nucleosome positioning.

2. Explain in detail "coupled dinucleotide contents"

3. Authors should do a rigorous literature survey on DNA structure and chromatin organization and interpret their results accordingly.

4. Authors should compare their results on classical experimental/theoretical assignments of DNA flexibility based on RR, YR, RY, and YY (R=Purine, Y=Pyrimidine). For instance, CA/TG and CG, are the most flexible, and RR steps, such as AA(=TT), are the most rigid.

Author Response

Thank you very much for the revision of our manuscript and your constructive feedback. We tried to consider each of your comments and resubmitted a corrected version. See below for answers on individual comments.

(1) Manuscript is very hard to understand, throughout the manuscript Authors used long sentences which are hard to grasp the idea.

We went through every single paragraph of the main text and tried to find clearer formulations for our statements and conclusions. Especially, we tried to use several shorter sentences instead of one overly long sentence, if possible.  We believe that our corrections improved readability and understandability considerably. Of course, our corrections also include the sentence you explicitly mentioned in your comment.

(2) Explain in detail "coupled dinucleotide contents"

We clarified the definition of “coupled dinucleotide contents”. Especially, we added a statement that we mean an evolutionary coupling, observable as a correlation of dinucleotide contents in (distant) Eukaryotes (91-93). We speak of a “coupling” and not simple of “conservation”, because the individual contents change (are not conserved) but the relation between contents is conserved.

(3) Authors should do a rigorous literature survey on DNA structure and chromatin organization and interpret their results accordingly.

We agree, that the relevance of DNA structure and chromatin organization for our discussion requires a more detailed introduction to the topic. Therefore, we added a paragraph to the introduction section of the paper where we discuss the relation between histone occupancy, DNA sequence, transcription and 3D organization of chromatin (37-51). The new paragraph also includes several additional literature sources for further reading.

(4) Authors should compare their results on classical experimental/theoretical assignments of DNA flexibility based on RR, YR, RY, and YY (R=Purine, Y=Pyrimidine). For instance, CA/TG and CG, are the most flexible, and RR steps, such as AA(=TT), are the most rigid.

Thank you for pointing at potential references for interpreting and comparing our results. We did a small literature survey on this interesting topic and re-checked our results accordingly. We found that flexibility of DNA was directly included in our dataset (e.g. a model for persistence length - see supplement table S4). The results for these models were inconclusive and therefore not part of the analysis presented in the main text). Nevertheless, we added a statement on DNA flexibility with reference to table S4 to the main text (540-543). Together with other minor changes in wording, this addition might also clarify, that we are only implying an indirect effect of certain DNA properties on the flexibility of chromatin and not an effect of/on DNA flexibility. We also added a sentence on bare linker DNA (580-582) where a direct impact of DNA properties on chromatin conformation would be plausible (linker DNA was also part of the new paragraph in the introduction section).

Reviewer 2 Report

Dinucleotide is one of the critical markers for eukaryotic genomes, like CpG associated with gene expression, species significance, and some diseases. Therefore, studying the dinucleotide patterns is very important for understanding the DNA and gene concepts. The author uses the bioinformatic method to analyze 2478 chromosomes of 155 eukaryotic samples and conclude the coupled dinucleotides influence chromatin organization. The data and results are reliable and reasonable. I think this paper can provide a good sight for people to better understand the mechanism for DNA property modification.  Thanks!

Author Response

Thank you very much for your encouragement and positive comments on our work.

Round 2

Reviewer 1 Report

The authors satisfactorily addressed my queries and incorporated them into the manuscript wherever necessary. I recommend the manuscript for publication. I request the Authors improve the readability of the manuscript.